# CASNET2: evaluation of an electronic safety netting cancer toolkit for the primary care electronic health record: protocol for a pragmatic stepped-wedge RCT

Susannah Fleming ![ORCID],[1] Brian D Nicholson ![ORCID],[1] Afsana Bhuiya,[2] Simon de Lusignan ![ORCID],[1,3,4] Yasemin Hirst,[5] Richard Hobbs,[1] Rafael Perera,[1] Julian Sherlock,[1,3] Ivelina Yonova,[1,3,4] Clare Bankhead[1]

For numbered affiliations see end of article.

**Correspondence to**
Dr Brian D Nicholson;
brian.nicholson@phc.ox.ac.uk

## ABSTRACT

**Introduction** Safety-netting in primary care is the best practice in cancer diagnosis, ensuring that patients are followed up until symptoms are explained or have resolved. Currently, clinicians use haphazard manual solutions. The ubiquitous use of electronic health records provides an opportunity to standardise safety-netting practices.

A new electronic safety-netting toolkit has been introduced to provide systematic ways to track and follow up patients. We will evaluate the effectiveness of this toolkit, which is embedded in a major primary care clinical system in England:Egerton Medical Information System(EMIS)-Web.

**Methods and analysis** We will conduct a stepped-wedge cluster RCT in 60 general practices within the RCGP Research and Surveillance Centre (RSC) network. Groups of 10 practices will be randomised into the active phase at 2-monthly intervals over 12 months. All practices will be activated for at least 2 months. The primary outcome is the primary care interval measured as days between the first recorded symptom of cancer (within the year prior to diagnosis) and the subsequent referral to secondary care. Other outcomes include referrals rates and rates of direct access cancer investigation.

Analysis of the clustered stepped-wedge design will model associations using a fixed effect for intervention condition of the cluster at each time step, a fixed effect for time and other covariates, and then include a random effect for practice and for patient to account for correlation between observations from the same centre and from the same participant.

**Ethics and dissemination** Ethical approval has been obtained from the North West—Greater Manchester West National Health Service Research Ethics Committee (REC Reference 19/NW/0692). Results will be disseminated in peer-reviewed journals and conferences, and sent to participating practices. They will be published on the University of Oxford Nuffield Department of Primary Care and RCGP RSC websites.

**Trial registration number** ISRCTN15913081; Pre-results.

## Strengths and limitations of this study

► The safety-netting toolkit being evaluated is already implemented and integrated in the Egerton Medical Information System (EMIS) Web, a brand of electronic health record (EHR), but is rarely used.

► Data collection will be carried out primarily through existing automated links between the practice EHR and the RCGP Research and Surveillance Centre, surveillance system, thereby limiting the additional research workload for participating practices.

► We are reliant on the cancer diagnosis, generally made in specialist care, being recorded in the primary care EHR system in a consistent and timely manner.

## INTRODUCTION
### Background

Safety-netting is regarded as the 'best practice' in cancer diagnosis in primary care.[1] It aims to ensure patients do not drop through the healthcare net but are followed up until symptoms are explained.[2] Our research highlights an absence of evidence on how best to safety-net, especially with patients with non-specific cancer symptoms.[1] Expert consensus, international survey data and interviews with general practitioners (GPs) and patients show that effective patient communication, shared decision making and improved clinical systems are needed to ensure that tests and referrals are followed up and recurrent consultations are identified in patients with unexplained symptoms.[3–5] To achieve this, significant improvements in electronic health record (EHR) utilisation, particularly data quality, are required, by integrating information and communication technology with clinical care.[6–8]

While there are clear guidance and recommendations for safety netting in primary care,[9] successful implementation of these recommendations rely on resources available at general practices. Universally accessible National Health Service (NHS) fail safes do not exist to ensure tests are conducted, returned and reconciled.[10] Confusion exists about which staff member is responsible for test communication.[11] Patients can be unaware of their responsibility to follow up investigations and referrals, assuming 'no news is good news,' and taking no action if they do not feel better or develop new symptoms.[12 13] The success of a systems-based approach to safety-netting is jeopardised by inadequate administrative processes and marked variation in approaches to follow up.[14] EHR-based interventions show promise: trials in the USA of electronic prompts increased the proportion of patients with cancer symptoms who receive follow-up.[8 15–17] However, despite reporting enthusiasm for new initiatives, GPs do not always engage with new information technology, and this driven in part by social and technical factors, such as pop-up fatigue and information overload and being under-resourced.[13 14 18–21]

An electronic safety-netting toolkit (E-SN toolkit) has been developed through consultation with GPs in the UK and was embedded within one of the major clinical systems in England—Egerton Medical Information System (EMIS) Web in May 2018. The toolkit is designed to replace existing verbal or paper methods of safety netting by providing effective means for tracking and follow up by administrative staff. The E-SN toolkit proposes a rigorous, robust, traceable and auditable proactive approach to tracking patients. It is designed to allow clinical data to be entered using templates, and diary entries to be generated (time reminders to check an action has been completed). Using diary entries, users can effectively follow up test requests, referrals and non-specific but concerning symptoms. For instance, outstanding actions appear as 'Alert Flags' to identify incomplete diary entries and can be collated. Although the E-SN toolkit is currently available to general practices using EMIS, practices have to proactively turn it on to implement it in their current practice.

This study aims to evaluate the effectiveness of an EHR safety-netting toolkit (E-SN toolkit) for use in primary care with patients with symptoms of cancer.

### Study design
The study will employ a stepped-wedge cluster-randomised controlled trial (SW-CRCT) randomising clusters, that is, general practices, in blocks of 10 to the timing of activation of the E-SN toolkit. General practices will crossover in these blocks to the activated phase every 2 months (figure 1). The study will compare patient's primary care interval data for cancer diagnoses pre and post E-SN toolkit activation at all participating practices. The stepped-wedge design will ensure that time-related confounders such as seasonal variation should be accounted for.

| Practices* | Pre-randomisation period (months) | | | | | | Post-randomisation cross-over period (months) | | | | | |
| --- | --- | --- | --- | --- | --- | --- | --- | --- | --- | --- | --- | --- |
| | -12 | -10 | -8 | -6 | -4 | -2 | 0-2 | 2-4 | 4-6 | 6-8 | 8-10 | 10-12 |
| 1-10 | | | | | | | | | | | | |
| 11-20 | | | | | | | | | | | | |
| 21-30 | | | | | | | | | | | | |
| 31-40 | | | | | | | | | | | | |
| 41-50 | | | | | | | | | | | | |
| 51-60 | | | | | | | | | | | | |

*Practices will be randomly allocated to the group and date of cross-over

**Figure 1** Stepped-wedge design with 12 months prerandomisation period. Pale blue cells represent inactive E-SN period, and purple cells represent active E-SN period. E-SN, electronic safety-netting.

This paper describes the protocol dated 7 October 2019 (V.1.5). Core trial information is given in table 1.

The anticipated length of the study is 18 months. This consists of 3 months recruitment, followed by a 12-month period during which time the intervention will be introduced, and 3 months for analysis.

## METHODS AND ANALYSES
### Outcomes and outcome definitions
#### Primary outcome: primary care interval for cancer diagnoses
The primary care interval is defined as the number of days between the first recorded symptoms of cancer (within the year prior to diagnosis) and subsequent referral for secondary cancer care.[22] In line with published research and guidelines on diagnostic intervals, we will search the patient record for coded symptoms during the year prior to diagnosis for all patients with a cancer diagnosis: 1 year is a trade-off between misattributing unrelated symptoms occurring more than a year before and missing symptoms of relevance by restricting to a shorter period.[22 23]

#### Secondary outcomes
Full details of the secondary outcomes are given in table 2 and figure 2.

### Setting and participants
The study will be carried out in 60 English general practices that contribute data to the Royal College of General Practitioners (RCGP) Research and Surveillance Centre (RSC) Network[24 25] and use the EMIS Web EHR system. The RCGP RSC includes general practices in England. A full list of NHS Clinical Research Networks from which practices will be recruited may be found in online supplementary appendix A.

#### Inclusion and exclusion criteria
The inclusion criteria for general practices are as follows:
► Practice is actively contributing data to the RCGP RSC.
► Utilises EMIS EHR system
► Data available for the previous 24 months.
The exclusion criteria for general practices are:
► Practices that express an interest, but are not fully set up for data extraction.
► Any practice already deploying the E-SN toolkit.

**Table 1** WHO trial registration data set

| Data category | Information |
|---|---|
| Primary Registry and Trial Identifying No | ISRCTN: ISRCTN15913081 |
| Date of Registration in Primary Registry | 08/11/2019 |
| Secondary Identifying Numbers | N/A |
| Source(s) of Monetary or Material Support | Cancer Research UK; Grant Codes: C48270/A27880 |
| Primary Sponsor | University of Oxford |
| **Secondary Sponsor(s)** | None |
| Contact for Public Queries | susannah.fleming@phc.ox.ac.uk |
| Contact for Scientific Queries | clare.bankhead@phc.ox.ac.uk |
| Public Title | Testing an electronic safety netting system to help GPs follow-up patients with worrying symptoms |
| Scientific Title | CASNET2: Evaluation of an e-safety netting cancer template in primary care: a pragmatic stepped-wedge RCT |
| Countries of Recruitment | UK |
| Health Condition(s) or Problem(s) Studied | E-safety netting (E-SN) toolkit. |
| Intervention(s) | The researchers will recruit 60 general practices who are not currently using the E-SN toolkit, and randomise them in clusters (groups) of 10. Each cluster will have the E-SN toolkit turned on at a different time during the 12 months of the study. Once the E-SN toolkit is turned on, the GPs in the practice will be able to use it when caring for any patient they think would benefit from it, although it is expected that it will be of most use when treating patients with symptoms that might indicate cancer. The researchers will collect data from the electronic patient record system from the 12 months of the study and the 24 months before the start of the study to understand whether the introduction of the E-SN toolkit makes any difference to the diagnosis of cancer, and in particular to how quickly patients are diagnosed. The researchers will only extract records from patients who are over 18, and who have not opted out of the research. |
| Key Inclusion and Exclusion Criteria | GP practices will be eligible for inclusion under the following conditions:<br>1. They are actively contributing to the RCGP Research and Surveillance Centre database.<br>2. They use the EMIS electronic health record system.<br>3. They have data available for the previous 24 months.<br>Within the participating practices, the researchers will seek to extract data from adult patients (aged over 18 years)<br>Exclusion criteria:<br>1. GP practices who are already using the E-SN toolkit will not be eligible for the study.<br>2. The researchers will not extract data from any patient under 18, or from any patient who has opted out of data sharing for research purposes. |
| Study type | Other |
| Date of First Enrolment | 25/11/2019 |
| Target Sample Size | 60 |
| Recruitment Status | Not yet recruiting |
| Primary Outcome(s) | Primary care interval for cancer diagnoses measured as the time between the first recorded symptom of cancer and referral to secondary cancer care, during inactive and active E-SN phases. |

Continued

**Table 1** Continued

| Data category | Information |
|---|---|
| Key Secondary Outcomes | 1. Proportion of cancers diagnosed after emergency presentation measured during inactive and active E-SN phases. |
| | 2. Recorded new diagnoses in those who have a template activated, measured by cancer site and stage, and by non-cancer diagnosis, during the active E-SN stage. |
| | 3. Total time to diagnosis measured from first recorded symptom to definitive diagnosis for all cancer diagnoses during the inactive and active E-SN phases and all diagnoses with template activation during the active E-SN phase |
| | 4. No of GP consultations/patient between first record of symptom and cancer referral, measured during the inactive and active E-SN phase |
| | 5. Rates of patients completing direct access cancer investigations measured during the inactive and active E-SN phase |
| | 6. Rates of patients referred measured as 2-week wait, urgent, and routine, during the inactive and active E-SN phase |
| | 7. Timing of template activation within the primary care interval (from first symptom to referral) measured during the active E-SN phase |
| | 8. Template activation rate among consulting patients, both total and stratified by individual GP, measured during the active E-SN phase |
| | 9. The proportion of diary entries completed measured during the active E-SN phase |
| | 10. The reason for template activation measured based on 20 high-level READ codes during the active E-SN phase |
| | 11. Symptoms leading to direct access to investigations measured during the active E-SN phase |
| | 12. Recorded vague symptoms in the template measured during the active E-SN phase |
| | 13. Demographic details of patients with activated templates measured during the active E-SN phase |
| | 14. GP type completing template (eg, partner, locum, trainee) measured during the active E-SN phase |
| | 15. Diagnostic codes in patients with activated templates measured during the active E-SN phase |

EMIS, Egerton Medical Information System; E-SN, electronic safety-netting; GP, general practitioner.

All patients over the age of 18 at participating practices are eligible for inclusion in the study, unless they have opted out of data sharing.

### Recruitment

The RCGP RSC network will identify potentially eligible practices and circulate the details of the study before and during the recruitment period. Practices will also be approached directly by the RCGP RSC Practice Liaison Officers (PLOs), who regularly visit practices to monitor their data quality and inform about current open studies. Expressions of interest will be obtained from all interested practices. An additional 12 practices will be identified to account for potential drop-outs during the study. All practices will be recruited at the start of the study prior to randomisation and implementation of the intervention.

General practices will receive reimbursement of up to £500 per practice for participation in the 12-month stepped-wedge intervention study.

### Randomisation and blinding

The eligible practices will be ranked according to their list size from smallest to the largest by the RCGP RSC. These will then be stratified into 10 strata (by list size) such that each strata contains the same number of practices. This is based on the allocation of 10 practices per step. Practices will be randomised in blocks of 10 to the timing of activation of the E-SN toolkit and will crossover in these blocks to the activated phase every 2 months. Therefore, practices will contribute between 2 months and 12 months of E-SN toolkit-activated time (figure 1).

**Table 2** Outcomes, measures and time periods of measurement for primary and secondary outcomes

| Outcome | Measure | Inactive study period | Active study period | Template activations only |
|---|---|---|---|---|
| **Primary outcome** | | | | |
| Primary care interval for cancer diagnoses | Measured as the no of days between first recorded symptoms of cancer (within the year prior to diagnosis) and subsequent referral for secondary cancer care | X | X | |
| **Secondary outcomes** | | | | |
| Proportion of cancers diagnosed after emergency presentation | Proportion of cancers for which diagnosis is made prior to referral, including following A&E or inpatient episodes. Where there is uncertainty regarding the route of diagnosis, the RCGP RSC network will contact the practice in an attempt to augment the data. Algorithms will also be developed to identify emergency presentations of cancer. | X | X | |
| Recorded new diagnoses in those who have a template activated | By cancer site and stage, and non-cancer diagnoses. Coded entries for all alternative diagnoses where the E-SN toolkit has been activated will be identified. | | X | X |
| Total time to diagnosis (from first recorded symptom to definitive diagnosis) | Measured from first recorded symptom of cancer (within the previous year) to definitive diagnosis for all cancers diagnosed, and for all patients with an activated template | X | X | X (for non-cancer diagnoses) |
| No of GP consultations/patient between first record of symptom and cancer referral | Measured as no of primary care consultations between the first recorded symptoms (within the year prior to diagnosis) and subsequent referral, per patient. | X | X | |
| Rates of patients completing direct access cancer investigations | Measured as the no of patients undergoing direct access cancer investigations (according to those specified in referral guidelines NG12) in each period divided by the person years of observation for that period. | X | X | |
| Rates of patients referred for cancer | Referrals rates via 2-week wait, urgent, and routine routes for all patients referred for specialist opinion to a secondary care cancer specialist | X | X | |
| Timing of template activation | Measured as the no of days between first recorded symptoms (within the year prior to diagnosis), and template activation and the no of days between template activation and subsequent referral. | | X | X |

**Table 2** Continued

| Outcome | Measure | Inactive study period | Active study period | Template activations only |
|---|---|---|---|---|
| Template activation rate among consulting patients | Measured as the number of patients with an activated template divided by the no of patients consulting, in each time period.<br>Both total rate and rate stratified by individual GP will be measured. | | X | X |
| Proportion of diary entries that were completed | Measured as the no of diary entries that were completed divided by the number of diary entries that were opened. | | X | X |
| Reason for template activation | The coded reasons for activating the template,<br>Based on 20 high level READ codes | | X | X |
| Symptoms leading to direct access to investigation | All symptoms recorded in patients undergoing direct access cancer investigations | X | X | |
| Recorded vague symptoms in the template | All symptoms recorded within the template. | | X | X |
| Demographic details of patients with activated templates | Age and sex of patients that had a template activated during the course of the trial. | | X | X |
| GP type completing templates | Descriptive data on the type of GP that first activated the template (eg, partner, locum, trainee) | | X | X |
| Diagnostic codes in patients with activated templates | Diagnoses recorded after the activation of template. | | X | X |

E-SN, electronic safety-netting; GP, general practitioner; RSC, Research and Surveillance Centre; RSC, Research and Surveillance Centre.

A statistician (Rafael Perera, Nuffield Department of Primary Care Health Sciences, University of Oxford) who is independent to the intervention development and implementation will produce a stratified randomisation schedule so that within each strata, practices are randomly allocated to each of the six steps, with some replacement practices. The random sequence will be generated using R software. The allocation will be undertaken for all practices at the same time. Where replacement practices are required, these will be taken from the same strata where possible. If no replacement practices from the same strata are available a coin toss will determine whether the next highest or lowest strata will be used to provide replacement practices.

Given the practice change nature of the intervention, clinicians and practice managers will be aware when their practice has switched to the intervention period. Consulting patients providing outcome data will not be informed of the experimental nature of the E-SN toolkit activation and therefore will be blind to the stage of study occurring in the practice they attended. Study personnel involved in extracting outcome data will be blind to the allocated order of the delivery of the intervention across the practices. All data management of extracted data to calculate the outcome measures will be conducted blinded to the timing of switching to intervention. Similar methods have been used in other implementation SW-CRCTs.[26]

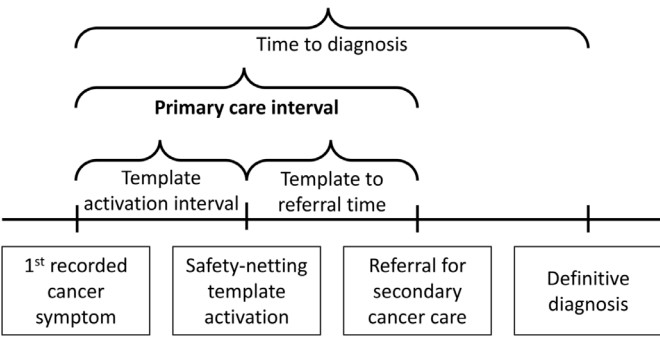

**Figure 2** Diagram showing the definitions of time-based outcomes.

### Intervention
The intervention consists of activation and the implementation of the E-SN toolkit at participating practices. All

practices will receive training in the use of the E-SN toolkit from RCGP PSC PLOs prior to their switching date. Clinicians will be able to use the E-SN templates at their own discretion during the active period, with no requirements on which patients should or should not receive safety netting. Clinical care of patients will continue as normal throughout the study.

The E-SN toolkit uses a templates to track cancer events like referrals, direct access tests and monitoring of low risk through read codes attached to diary entries. Expired diary entries would be identified using automated searches and actioned by the administration lead as appropriate. If an event was complete, for example, normal scan results done in 2 weeks and result back—then the diary entry is closed, resolving the episode. The E-SN toolkit has extra features such as popup alerts to remind any user there is an open diary entry and also allows the E-SN toolkit to popup automatically if a relevant code is used in the clinical records.

Further details and demonstration video are available from: https://www.uclh.nhs.uk/OurServices/ServiceA-Z/Cancer/NCV/MICa/Pages/Primarycareimprovement.aspx

Practices will all be encouraged to adhere to the schedule for switching and will receive part-payment for study initiation and then full payment when they adhere to the activation schedule.

### Data extraction

Data extractions from all participating practices will correspond to two major time points: at the start of the introduction to the stepped wedge implementation of the E-SN toolkit, and at the end of the stepped wedge period (12 months later). At these two time points, consultation data from the participating practices will be obtained for the previous 24 months. Data from the previous 24 months are required to identify cancers detected in 12 months prior to start, plus the first recorded symptom of cancer during the primary care interval, allowing for this interval to cover up to a year prior to diagnosis.

Records from all patients who are adults aged 18 years and above, registered at participating practices and who have not opted out of sharing data for research purposes will be extracted. Pseudonymised patient information will include: demographic information of age, sex, dates of clinical consultations; coded consultation data for symptoms, diagnoses, tests ordered and referrals made within the consultation records. Where a definitive diagnosis has been made (for cancer and non-cancer conditions), the clinical features recorded in the year prior to diagnosis will be captured. These actions will be achieved by extraction of the EHR rather than by hand searching notes.

Practices will be asked to search for open diary entries every week during the intervention period, and to download and save these to an Excel spreadsheet (locally). The central study research team will use these spreadsheets to track which diary entries are closed during the course of the intervention.

### Data management

The principal data source for the this study are pseudonymised routinely collected care data extracted from general practices of the RCGP RSC network's database; there are over 500 general practices in this network. All general practices in the UK use an EHR to maintain patient medical records. Data are entered into a patient's computerised medical system as coded data or free text. The RCGP RSC extract only coded data, that is, where the clinician codes a disease or symptom into their EHR system.[27]

The RCGP RSC has no role in updating clinical data recorded by clinicians as part of their consultation and care. However, the RCGP RSC practices do received focused feedback about data quality around its surveillance function, influenza other infections, vaccine update and effectiveness. This is done via team of PLOs and a dashboard.[28 29]

RCGP RSC maintains an auditable trail for all the stages of data processing to ensure the quality of data are not compromised by the processing. For example, checking the prevalence of certain conditions and outliers revealed by the data is consistent with those reported in the literature. The standard operating procedures for data extraction and data processing, and data access are available from https://clininf.eu/index.php/information-governance/

### Discontinuations and withdrawals from the study

Each practice has the right to withdraw from the study at any time. Data from withdrawn practices will be included in analyses up to the point of withdrawal, unless they indicate that they wish to withdraw previously collected data from analysis.

### Sample size

Practice lists sizes within the RCGP RSC are approximately 10 000. In England, diagnosis rate of new cancer was 523/100 000 per year (2014/2015).[30] Therefore, we could expect 53 new cancers per year per practice. Therefore, in each 2-month step, there would be 8–9 cancers per cluster.

The median primary care interval between first presentation and specialist referral[22 31] is 5 days, IQR of 0–27.[32] Some cancers present with clear red flag symptoms leading to immediate specialist referral. Presentations of vague symptoms such as weight loss are less likely to be immediately referred and may benefit from using the safety-netting template. This symptom is associated with several cancers such as prostate, colorectal, lung, gastro-oesophageal and pancreatic.[33] The median primary care interval for lung cancer is 14 days (IQR 2–45).[32] Using Stata V.14 to conduct the sample size calculations we showed that with the design in figure 1 and 60 practices

**Table 3** Scenarios for various assumptions used in the sample size calculations

| Assumptions | All cancers (based on lung cancer) | All cancers | All cancers | All cancers | Restricting to only 90th centile delays | Restricting to only 90th centile delays |
|---|---|---|---|---|---|---|
| Median (days) | 14 | 14 | 60 | 60 | 60 | 60 |
| Range (days) | 0–60 | 0–100 | 0–365 | 0–100 | 14–100 | 14–365 |
| N of cancers per step per cluster | 9 | 9 | 9 | 9 | 1 | 1 |
| Minimum detectable difference (days) | 2 | 5 | 13 | 4 | 9 | 39 |
| Notes | | Allowing greater range | | Lower upper value for Primary Care Interval | Minimum set to median of all cancers | Min set to median of all cancers, but increased upper range |

we would be able to detect an effect size of 2 days with 80% power.

Currently, approximately 19% of cancers are diagnosed following an emergency presentation.[34] With nine cancers per step per cluster we would be powered to detect a difference of 5%.

Under another scenario of considering primary care intervals towards the 90th centile of 60 days, with 60 practices, entering the stepped wedge design in six steps we would be able to detect a minimal difference of 13 days. However, if we consider that these patients with longer delays are in the 90th centile, then instead of expected cancers per cluster per step of 9—there would be around 1. This would allow us to detect a minimal difference of between 9 and 39 days dependent on the assumption of the distribution of the primary care delays.

Several scenarios are shown in table 3, all based on 60 practices, entering in six blocks, with a 12-month preintervention period. In summary, our main analysis will focus on all cancers, but we can conduct prespecified subgroup analyses restricting to cancers that typically have longer delays.

### Statistical analysis

In patients with a diagnosis of cancer, we will calculate the primary outcome of primary care interval (time between first recorded symptom and referral), proportion diagnosed after emergency presentation and the recorded diagnosis. Among consulting patients, we will calculate the rate of direct access cancer investigation and rates of referrals via 2 weeks wait, urgent and routine pathways, as well as the number of consultations during the primary care interval, and the template activation rate. A patient referred to: a 2-week wait pathway will be reviewed by a specialist within 2 weeks; an urgent pathway will be reviewed as a priority but the exact timing varies between specialty and healthcare setting; a routine pathway requires specialist review but on a non-urgent basis.

Regarding the analysis of the stepped-wedge design and the effect of correlation of observations within clusters, will model the association using a fixed effect for the intervention condition of the cluster at each time step, a fixed effect for time and other covariates (eg, changes in cancer guidelines), and then include a random effect for practice and a random effect for patient to account for correlation of the observations from the same centre and from the same participant.

Analyses will include all patients registered at participating practices. Practices that withdraw their agreement to participate will be included in analyses up to the point of withdrawal, unless they indicate that they wish to withdraw previously collected data from analysis.

Our primary analysis will be carried out on an intention to treat principle. Therefore, if any practice does not switch on the E-SN toolkit at the correct time for their cluster, we will carry out the analysis under the assumption that the E-SN toolkit was switched on at the correct time.

The primary care interval is defined from the presence of symptoms and referral in the primary care record. Symptoms of interest will include all symptoms included in the urgent referral guidelines including vague symptoms such as weight loss, tiredness and back pain.[35 36] Missing data for these variables in the record will be interpreted as the absence of symptoms or referral, respectively. Extracted data from the primary care record will be limited to the study period plus the preceding year, so symptoms recorded prior to this will not be visible to the research team.

The study will not have a formal data monitoring committee as patients will be receiving standard care through their GP, and all data will be extracted from routinely collected clinical notes.

### Planned subgroups

Where applicable, subgroups will be:
▶ Patients in whom an E-SN toolkit entry was completed.
▶ Patients diagnosed with cancer.

### Data display and reporting

We will combine or suppress any cells with small numbers (under 5) of observations to prevent any potential identification during the reporting of the results.

## Patient and public involvement

We have recruited an experienced patient and public involvement (PPI) representative to sit on the project steering committee. She has been involved in planning how PPI involvement will best fit in with the project, and will attend steering committee meetings to discuss project progress, as well as cochairing workshops with patient representatives. We anticipate involving five additional patient representatives. The PPI collaborators will be involved in interpreting the findings of the study and identifying which of the prespecified outcomes are of greatest importance to patients. This will allow them to consider whether there would be a beneficial effect in terms of patient perspectives, even if the E-SN toolkit caused only a small change in outcomes.

## ETHICS AND DISSEMINATION
### Informed consent

Practices that are randomised to take part in the study will be provided with a welcome pack by their RCGP RSC PLO team. The pack includes a practice information leaflet (online supplementary appendix B) and practice poster (online supplementary appendix C), a copy of the protocol, and a copy of the ethical approval documentation, providing the practice with detailed information about the exact nature of the study; study requirements; the implications and constraints of the protocol; and any risks involved in taking part. Site agreements will be in place with each practice and it will be clearly stated that the practice is free to withdraw from the study at any time for any reason, without affecting their legal rights, and with no obligation to give the reason for withdrawal. Signing a site agreement will form consent for a practice to take part in the study.

No direct or active involvement will be required from consulting patients and we will not be seeking individual patient consent. The rationale for obtaining agreement at the cluster (practice) level is that the activation of the Toolkit will be through the EMIS Web software system and healthcare practitioners are the intended recipient of the intervention.[37 38]

Patients who have opted out of sharing data are not processed by the RCGP RSC and therefore will not be accessed the research team and their data will not be extracted. RCGP RSC does keep a count of opt-out patients per practice as this is needed to interpret rates collected for surveillance, this runs at around 2% of the registered population.

A patient notification in the form of a poster will be displayed in all participating practices giving patients information about the study and how to opt out of data sharing. Outcome data will be extracted from coded data in the EHR by the SQL developer and provided in a pseudonymised form to the analysis team.

### Safety reporting

As patients remain under their GP's care throughout the study, and serious adverse events such as death and hospitalisation unrelated to the study are expected in this patient group, no formal monitoring of serious adverse events will be carried out.

### Quality assurance

The study may be monitored, or audited in accordance with the current approved protocol, Good Clinical Practice (GCP), relevant regulations and standard operating procedures by responsible individuals from the sponsor and the NHS trusts in which it is being carried out.

### Ethical and regulatory considerations

Ethical approval has been obtained from the North West—Greater Manchester West NHS Research Ethics Committee (REC Reference 19/NW/0692). We have also obtained Health Research Authority (HRA) approval to carry out the study in the NHS. The study sponsor is the University of Oxford, UK (ctrg@admin.ox.ac.uk). All substantial amendments to the protocol will be submitted to the sponsor, the ethics committee, and the HRA, and, where necessary, their approval will be obtained.

The investigator will ensure that this study is conducted in accordance with relevant regulations and with GCP. The university has a specialist insurance policy in place which would operate in the event of any participant suffering harm as a result of their involvement in the research (Newline Underwriting Management, at Lloyd's of London). NHS indemnity operates in respect of the clinical treatment that is provided. All participants will continue to receive NHS care during and after the study.

### Patient confidentiality

The study staff will ensure that the practices' patients' anonymity is maintained. The practice patients will be identified only by an ID number on all study documents and any electronic database.

All documents will be stored securely and only accessible by study staff and authorised personnel. The study will comply with the UK General Data Protection Regulation and Data Protection Act 2018, which require data to be anonymised as soon as it is practical to do so. Pseudonymisation will be carried out using the RCGP RSC standard processes and will ensure that it is not possible for research staff to link study data with data from other sources. Although multiple items will be extracted from individual clinical records, we have taken care to minimise the number of data items/variables made available for this analysis, for example, using age rather than date of birth. RCGP RSC apply a process of statistical disclosure control to ensure that individuals cannot be identified, even from aggregate data. For example, data might have to be exported by 5-year or 10-year age bands if there were small numbers in an individual year of birth.

Data are held on a protected by a firewall secure server at the University of Surrey, currently acting as the RCGP RSC's data and analysis hub. All in-bounded connections are blocked, but out-bounded connections are allowed on approval by a senior staff member.

**Table 4** Membership of groups overseeing the study

| Member | Role | Day-to-day management | Management group | Advisory board |
|---|---|---|---|---|
| Susannah Fleming | Study coordinator | Yes | Yes | Yes |
| Clare Bankhead | PI | Yes | Yes | Yes |
| Ivelina Yonova | Project manager | As necessary | Yes | Yes |
| Brian Nicholson | Co-PI | As necessary | Yes | Yes |
| Simon de Lusignan | Coinvestigator | | Yes | Yes |
| Yasemin Hirst | Coinvestigator | As necessary | Yes | |
| Afsana Bhuiya | GP—creator of E-SN toolkit | As necessary | Yes | |
| Rafael Perera | Statistician | As necessary | Yes | |
| Julian Sherlock | Programmer | As necessary | Yes | |
| Sue Duncombe | PPI representative | As necessary | Yes | Yes |
| Jodie Moffatt | Funder representative | | | Yes |
| Rebecca Canning-Johns | Independent statistician | | | Yes |
| Richard Hobbs | Coinvestigator | | Yes | |
| Kathy Pritchard-Jones | Advisor | | Yes | |

E-SN, electronic safety-netting; GP, general practitioner; PI, principal investigator.

The secure server is managed by the Clinical Informatics and Health Outcomes Research Group, and will be moved to the University of Oxford where the Group is based. It meets the requirements of NHS Digital's Data Security and Privacy toolkit.

### Study management

The study will be overseen by three groups (see table 4 for composition and roles of members.) Day-to-day management of the study will be primarily carried out by a core group of three researchers, meeting weekly to discuss any ongoing issues, with other members of the management group brought in to advise as necessary.

The study management group will meet quarterly and will provide pragmatic scientific and methodological support, management oversight, and will participate in dissemination activities and planning of future funding applications.

The study advisory board will meet quarterly and will focus on strategic oversight and progression reviews, and dissemination activities.

### Dissemination

The primary results of the study will be published in a peer-reviewed journal publication. Additional dissemination may take place via peer-reviewed conference presentations, and additional journal publications. Participating general practices will receive a report of the main findings, and results will also be disseminated on both the University of Oxford Nuffield Department of Primary Care and RCGP RSC websites. Patients will not be directly informed of the results, but will be able to access results on the internet if they so wish.

Authorship of all publications will be determined in accordance with the International Committee of Medical Journal Editors guidelines.

The final trial dataset will consist of a large quantity of pseudonymised participant-level data extracted from clinical records. Due to the potentially sensitive nature of this data, and to ensure compliance with relevant data protection law, this data will only be made accessible to the subgroup within the research team who will be carrying out the analysis, and there are no plans for granting public access to the data.

The full protocol will be made available as part of the study ISRCTN registration details following publication of this paper. Statistical code will be available from the authors on request.

### Cancer diagnosis in the time of COVID-19

The COVID-19 pandemic has led to considerable changes in general practice, including a switch from predominantly face-to-face consultations to remote consultations by telephone or video call. In addition, many non-urgent hospital outpatient consultations and investigations have been cancelled or delayed, and it is likely that patients are less willing to consult with clinicians for what they perceive as 'minor' conditions, due to a desire to reduce pressure on the health service. All of these factors are likely to impact on cancer diagnosis and referral.

As a result, the initiation of the CASNET2 study has been delayed in an attempt to minimise the effect of these factors on the results of the study, and to ensure that the results are generalisable. However, we also intend to carry out supplementary downloads from participating practices for the 12 months prior to March 2020 (baseline comparison period) and for the entire pandemic period, or until 31 October 2020, whichever is latest.

We will then conduct a before-and-after analysis of the following outcomes:

▶ Presenting symptoms of cancer.

- ► Primary care interval for cancer diagnosis.
- ► Proportion of cancers detected after emergency presentation.
- ► Site and stage of new cancers
- ► Total time to diagnosis.
- ► Number of consultations in primary care interval.
- ► Rates of patients completing direct access cancer investigations.
- ► Rates of patients referred by 2-week wait, urgent and routine routes.

**Author affiliations**
[1]Nuffield Department of Primary Care Health Sciences, University of Oxford, Oxford, UK
[2]North Central and East London Cancer Alliance, London, UK
[3]Department of Clinical and Experimental Medicine, University of Surrey, Guildford, UK
[4]Research and Surveillance Centre, Royal College of General Practitioners, London, UK
[5]Research Department of Behavioural Science and Health, University College, London, UK

**Contributors** CB, BDN, RH, RP, AB, YH, SdL, IY and JS contributed to writing the original grant application, on which the protocol was based. SF, CB and BDN wrote the original protocol, with contributions from RP, AB, IY and SdL. SF, CB and BDN wrote the paper. All authors commented on the paper and approved the final version.

**Funding** This work is supported by Cancer Research UK, Early Diagnosis Advisory Group (EDAG) grant number C48270/A27880. The CASNET2 COVID-19 extension (Cancer Diagnosis in the time of COVID-19) is funded by the University of Oxford Medical Sciences Division COVID-19 Research Response Fund (ref 0009070). CB is supported by the NIHR Oxford Biomedical Research Centre and the Applied Research Collaborative, Oxford. AB is funded by the NCEL Cancer Alliance. YH is funded by a Cancer Research UK Early Detection Grant (C38463/A26726). RH acknowledges his part-funding from the National Institute for Health Research (NIHR) School for Primary Care Research, the NIHR Collaboration for Leadership in Health Research and Care (CLARHC) Oxford, the NIHR Oxford Biomedical Research Centre (BRC), and the NIHR Oxford Medtech and In-Vitro Diagnostics Co-operative (MIC). SdL has received commercial funding through his university for research funded by Seqirus, GSK, Lilly, Novo Nordisk, Astra Zenaca and Takeda principally for unrelated research in the areas of diabetes and vaccination. RP also received funding from the National Institute for Health Research (NIHR) Oxford Biomedical Research Centre, the NIHR Oxford Medtech and In-Vitro Diagnostics Co-operative, the NIHR Applied Research Collaboration Oxford and Thames Valley, and the Oxford Martin School.

**Competing interests** AB is the innovator of the E-SN toolkit and locally promotes the toolkit for the purposes of cancer care along its pathway. SF is a member of the National Body Temperature Measurement Group. SdL is director of the RCGP, as part of his academic work.

**Patient and public involvement** Patients and/or the public were involved in the design, or conduct, or reporting, or dissemination plans of this research. Refer to the Methods section for further details.

**Patient consent for publication** Not required.

**Provenance and peer review** Not commissioned; externally peer reviewed.

**ORCID iDs**
Susannah Fleming http://orcid.org/0000-0001-7205-2051
Brian D Nicholson http://orcid.org/0000-0003-0661-7362
Simon de Lusignan http://orcid.org/0000-0001-5613-6810

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
