## [Reviewer comments · BMJ Open]

ARTICLE DETAILS

TITLE (PROVISIONAL)	CASNET2: Evaluation of an Electronic Safety Netting cancer toolkit for the primary care electronic health record: protocol for a pragmatic stepped-wedge RCT
AUTHORS	Fleming, Susannah; Nicholson, Brian; Bhuiya, Afsana; de Lusignan, Simon; Hirst, Yasemin; Hobbs, Richard; Perera, Rafael; Sherlock, Julian; Yonova, Ivelina; Bankhead, Clare

VERSION 1 – REVIEW

REVIEWER	Eric Thomas McGovern Medical School at the University of Texas Health Science Center Houston. Houston, Texas, USA
REVIEW RETURNED	24-Apr-2020

GENERAL COMMENTS	1. Page 7, lines 25-33, describe the primary care interval for cancer diagnoses. I find this confusing. The first sentence generally defines the interval, and that is clear. But the second sentence seems to be describing how patients will be identified (“we will search the patient record for all patients with a cancer diagnosis for the year prior to diagnosis....”) That does not seem to tell readers how the interval will be measured. The phrase “search the patient record for all patients with a cancer diagnosis for the year prior to diagnosis” is does not make sense. How does one search for a diagnosis prior to the diagnosis, when there is no diagnosis? Looking at Table one, I suggest including the definition in table 1 here in the text. 2. I'd encourage the authors to provide a brief explanation of the “steppedwedge, detectable difference incomplete” command in Stata. 3. This next comment may be related to the one above: I would have expected the sample size section to include an estimate of the effect size of the intervention.
---

REVIEWER	Claudia Lai PhD, post doctoral fellow France Légaré MD, PhD, supervisor Université Laval Canada
REVIEW RETURNED	14-May-2020

GENERAL COMMENTS	Please see below.
-------------------

Topic	Comment on paper	Questions
1. Rationale for step-wedge design	The authors stated that the stepped-wedge design “will ensure that time-related confounders such as seasonal variation should be accounted for.” (p7 line 12)	What is the rationale for using a cluster design and rationale for using a stepped wedge design rather than a simple parallel arm implementation?
2. Primary outcome	The primary outcome is defined as the interval between date of first recorded cancer symptoms (within the year prior to diagnosis) and subsequent referral to secondary care. The authors state that data “are entered into a patient’s computerised medical system as coded data or free text.” (Page 9 – line 50) The authors state that the outcome data “will be extracted from the HER by the SQL developer and provided in a pseudonymised form to the analysis team.” (page 12–line 22)	It is unclear how the SQL developer will determine date of first recorded cancer symptoms based on clinical data entered as free text.
3. Secondary outcomes	Other outcomes are defined as rates of referrals via the UK’s very urgent 2 week wait, urgent and routine pathways, and the rate of direct access cancer investigation.	These outcomes could be more clearly defined.
4. Addition outcomes as identified by patient and public involvement collaborators	The authors state that “PPI collaborators will be involved in interpreting the findings of the study and identifying outcomes that are of greatest importance to patients.” (page 11–line 47)	Will study outcomes be modified after protocol is published?
5. Clinical adoption of tool	The authors indicate that “all practices will receive training in the use of the E-SN toolkit” (page 8 - line 55) prior to their switch date. And, the practices will receive “full payment when they adhere to the activation schedule.” (page 9 - Line 16) However, it is also noted that clinicians “will be able to use the E-SN templates at their own discretion during the active period, with no requirements on which patients should or should not receive safety netting.” (page 8, line 57)	Will differences in adoption rates at the clinician level impact the measured outcomes?

REVIEWER	Debra Choi Baylor College of Medicine - Houston, Texas, United States
REVIEW RETURNED	18-May-2020

GENERAL COMMENTS	The authors who propose this study seek to evaluate the effectiveness of an EHR safety-netting toolkit in patients with symptoms of cancer in a primary care setting. The authors will use a stepped wedge cluster-randomized approach to evaluate differences in the diagnostic interval. Overall, this protocol is well written and the details of the proposed study are thoughtfully described. If possible, I would suggest addressing the following issues:
---

	1) Could you clarify your inclusion/exclusion criteria for patients? For example, will you exclude patients who have any other diagnosis for a terminal illness, had any recurrent cancers prior to the specified primary care interval, etc.? 2) Will you be targeting specific cancers or all cancers in your study? Identifying all cancers seems overwhelming given the huge variation between symptom signatures and cancers. How will you identify relevant symptoms, especially in patients who have comorbidities? 3) How will you account for patients with screen-detected cancers and length bias? Slow growing tumors may have a longer pre-symptomatic period and this may affect findings of the study. 4) Participating GP practices may differ from non-participating GP practices in important aspects of the diagnostic process. Does any comparative data exist to adjust for bias? 5) How has COVID-19 affected wait periods for urgent cancer referrals? Findings would be extremely interesting.
--	---

VERSION 1 – AUTHOR RESPONSE

Reviewer: 1

Reviewer Name: Eric Thomas

Institution and Country: McGovern Medical School at the University of Texas Health Science Center Houston. Houston, Texas, USA

Please state any competing interests or state 'None declared': None

1. Page 7, lines 25-33, describe the primary care interval for cancer diagnoses. I find this confusing. The first sentence generally defines the interval, and that is clear. But the second sentence seems to be describing how patients will be identified (“we will search the patient record for all patients with a cancer diagnosis for the year prior to diagnosis....”) That does not seem to tell readers how the interval will be measured. The phrase “search the patient record for all patients with a cancer diagnosis for the year prior to diagnosis” is does not make sense. How does one search for a diagnosis prior to the diagnosis, when there is no diagnosis? Looking at Table one, I suggest including the definition in table 1 here in the text.

Response: Thank you. We have clarified the text to read:

The primary care interval is defined as the number of days between the first recorded symptoms of cancer (within the year prior to diagnosis) and subsequent referral for secondary cancer care. In line with published research and guidelines on diagnostic intervals, we will search the patient record for coded symptoms during the year prior to diagnosis for all patients with a cancer diagnosis: one year is a trade-off between misattributing unrelated symptoms occurring more than a year before and missing symptoms of relevance by restricting to a shorter period.(22, 23)

2. I'd encourage the authors to provide a brief explanation of the “stepped wedge, detectable difference incomplete” command in Stata.

Response: Thanks. We have removed the reference to the command relating to the power calculation and have altered the text to read:

The median primary care interval for lung cancer is 14 days (interquartile range 2-45).(32) Using Stata 14 to conduct the sample size calculations we showed that with the design in Figure 1 and 60 practices we would be able to detect an effect size of 2 days with 80% power.

3. This next comment may be related to the one above: I would have expected the sample size section to include an estimate of the effect size of the intervention.

Response: This has been addressed in the point above

Reviewer: 2

Reviewer Name: Claudia Lai PhD, post doctoral fellow France Légaré MD, PhD, supervisor

Institution and Country: Université Laval, Canada

Please state any competing interests or state 'None declared': None declared

Topic	Comment on paper	Questions
1. Rationale for stepwedge design	The authors stated that the stepped-wedge design “will ensure that time-related confounders such as seasonal variation should be accounted for.” (p7 line 12)	What is the rationale for using a cluster design and rationale for using a stepped wedge design rather than a simple parallel arm implementation?
Response: A cluster design has been chosen as the eSN Toolkit will be switched on within the GP practices electronic patient record system and therefore will be available to the whole practice at the same time, we therefore felt that it was necessary to conduct a clustered RCT. The stepped wedge design was chosen as it is a recommended method for evaluating service delivery interventions such as the eSN, which are likely to be rolled out over time. This design is attractive to participating practices as by the end of the trial they have all been exposed to the intervention, but it is still conducted in a randomised fashion.		
2. Primary outcome	The primary outcome is defined as the interval between date of first recorded cancer symptoms (within the year prior to diagnosis) and subsequent referral to secondary care. The authors state that data “are entered into a patient’s computerised medical system as coded data or free text.” (Page 9 – line 50) The authors state that the outcome data “will be extracted from the HER by the	It is unclear how the SQL developer will determine date of first recorded cancer symptoms based on clinical data entered as free text.

	SQL developer and provided in a pseudonymised form to the analysis team.” (page 12–line 22)	
Response: only coded data will be interrogated to ascertain the outcomes for this study, including date of first recorded cancer symptoms. We have made minor edits to the text to clarify this. (pages 6 and 9)		
3. Secondary outcomes	Other outcomes are defined as rates of referrals via the UK’s very urgent 2 week wait, urgent and routine pathways, and the rate of direct access cancer investigation.	These outcomes could be more clearly defined.
Response: We’re sorry this was opaque. We have removed “very” from the description of the 2 week wait pathway and added following the sentence: A patient referred to: a 2 week wait pathway will be reviewed by a specialist within 2 weeks; an urgent pathway will be reviewed as a priority but the exact timing varies between speciality and healthcare setting; a routine pathway requires specialist review but on a non-urgent basis.		
4. Addition outcomes as identified by patient and public involvement collaborators	The authors state that “PPI collaborators will be involved in interpreting the findings of the study and identifying outcomes that are of greatest importance to patients.” (page 11– line 47)	Will study outcomes be modified after protocol is published?
Response: Study outcomes will not be modified after protocol publication. The PPI collaborators will be prioritising the pre-specified outcomes, rather than creating new outcomes. This has been clarified in the text. (page 8)		
5. Clinical adoption of tool	The authors indicate that “all practices will receive training in the use of the E-SN toolkit” (page 8 - line 55) prior to their switch date. And, the practices will receive “full payment when they adhere to the activation schedule.” (page 9 - Line 16) However, it is also noted that clinicians “will be able to use the E-SN templates at their own discretion during the active period, with no requirements on which patients should or should not receive safety netting.” (page 8, line 57)	Will differences in adoption rates at the clinician level impact the measured outcomes?

Response: We will be measuring differences in template activation rate stratified by individual GP (see Table 1), so that we can assess this.

Reviewer: 3

Reviewer Name: Debra Choi

Institution and Country: Baylor College of Medicine - Houston, Texas, United States

Please state any competing interests or state 'None declared': None declared

The authors who propose this study seek to evaluate the effectiveness of an EHR safety-netting toolkit in patients with symptoms of cancer in a primary care setting. The authors will use a stepped wedge cluster-randomized approach to evaluate differences in the diagnostic interval. Overall, this protocol is well written and the details of the proposed study are thoughtfully described.

If possible, I would suggest addressing the following issues:

1) Could you clarify your inclusion/exclusion criteria for patients? For example, will you exclude patients who have any other diagnosis for a terminal illness, had any recurrent cancers prior to the specified primary care interval, etc.?

Response: inclusion and exclusion criteria for patients have been made explicit (page 4)

2) Will you be targeting specific cancers or all cancers in your study? Identifying all cancers seems overwhelming given the huge variation between symptom signatures and cancers. How will you identify relevant symptoms, especially in patients who have comorbidities?

Response: we will be including all cancers in this study. The coded symptoms recorded in the medical records will be extracted for the 12 month period prior to the diagnosis. Symptoms of interest will include all symptoms included in the urgent referral guidelines including vague symptoms such as weight loss, tiredness, and back pain.

3) How will you account for patients with screen-detected cancers and length bias? Slow growing tumors may have a longer pre-symptomatic period and this may affect findings of the study.

Response: Screening is only available for breast cancer, cervical cancer and colorectal cancer in the UK for limited age range of the population. This project is focusing on symptomatic patients that present to primary care.

4) Participating GP practices may differ from non-participating GP practices in important aspects of the diagnostic process. Does any comparative data exist to adjust for bias?

Response: It is possible that practices that participate in research are different to practices who do not engage in research and that will affect the external validity of the results. We are currently undertaking a larger study which will include both participating and non-participating practices and would allow us to comment on the baseline characteristics of these practices.

5) How has COVID-19 affected wait periods for urgent cancer referrals? Findings would be extremely interesting.

Response: The manuscript was submitted before COVID-19 was a significant issue in the UK. However, in the meantime, we have developed and obtained funding for an extension to CASNET2 to investigate the effect of COVID-19 on cancer diagnosis and referral, and have detailed this at the end of the manuscript (page 11)

VERSION 2 – REVIEW

REVIEWER	Eric Thomas The McGovern Medical School at The University of Texas Health Science Center Houston. USA
REVIEW RETURNED	22-Jun-2020

GENERAL COMMENTS	No additional comments.
-------------------------

REVIEWER	Debra Choi Baylor College of Medicine - Houston, Texas, United States
REVIEW RETURNED	01-Jul-2020

GENERAL COMMENTS	The authors who propose this study seek to evaluate the effectiveness of an EHR safety-netting toolkit in patients with symptoms of cancer in a primary care setting. The authors will use a stepped wedge cluster-randomized approach to evaluate differences in the diagnostic interval. Overall, this protocol is well written and the details of the proposed study are thoughtfully described. I would suggest creating a table/document specific to your study noting all the cancers with corresponding symptoms - if possible.
---